

**Comment on "Growth responses of trees and understory plants to**
**nitrogen fertilization in a subtropical forest in China" by Tian et al.**
**(2017)**
Taiki Mori[1]
[1]Kyoto University, Kitashirakawa, Sakyo-ku, Kyoto, Japan

**Abstract**
Negative effects of over-fertilization have been long reported in agricultural field, which
is known as fertilizer burn. A recent paper by Tian et al. (2017) reported a result of
simulated nitrogen (N) deposition experiment and demonstrated that application of
$NH_4NO_3$ solution significantly reduced small trees, understory saplings, shrubs, seedlings,
and ferns, while large trees were not affected by the application. They discussed that the
result was due to the reduced light availability and intensified N saturation. I challenge
this view, because it is more likely that the negative effects were caused by the monthly
application of $NH_4NO_3$ solution with high concentration (as high as 0.4 $M$ and 0.8 $M$).
Since experiments using liquid $NH_4NO_3$ are common, careful interpretations are also
required for other experiments.




*Key words: fertilizer burn; nitrogen deposition; nitrogen fertilization; nitrogen saturation*

**Text**
For testing the impacts of elevated nitrogen (N) deposition on ecosystems, including the
impacts on forest understory, plenty of manipulating experiments have been performed,
some of which applied high concentration of $NH_4NO_3$ solution as N source. A recently-
published paper by Tian et al. (2017) is one of them. They reported a remarkable negative
effect of $NH_4NO_3$ application on small-sized plants including trees, understory saplings,
shrubs/seedlings, and ferns, while the effect on large trees was not clear. Tian et al. (2017)
attributed the result to the reduced light availability and intensified N saturation.

However, I suspect that the negative impact on understory observed by Tian et

al. (2017) was due to the high concentration of the added N solution. Nitrogen is one of
the most important nutrients for plants, and often applied as a fertilizer in agricultural
practices. However, too much usage of the fertilizer can damage or even kill plants, which
has been known as "fertilizer burn." In the case of the Tian et al (2017)'s experiment, it
is likely that the high concentration of $NH_4NO_3$ solution caused foliar fertilizer damage
(Neumann *et al.*, 1981), reducing understory vegetation. The $NH_4NO_3$ solution applied



by authors were around 0.4 $M$ and 0.8 $M$ (0.48 and 0.95 kg $NH_4NO_3$ dissolved in 15L of
fresh water) in N50 (50 kg N $ha^{-1}$ $yr^{-1}$), and N100 (100 kg N $ha^{-1}$ $yr^{-1}$) sites, respectively
(materials and methods 2.1 in their paper). According to Neumann *et al.* (1981)'s
experiment, the concentration at which 20 µl droplets of $NH_4NO_3$ solutions applied to
leaf surface began to induce damage was 0.40 $M$. Therefore, it is very natural to assume
that monthly application of 0.4 $M$ and 0.8 $M$ $NH_4NO_3$ solution can damage forest
understory.
Authors tried to explain the decrease in understory vegetation in several parts of
the manuscript, but their hypotheses seem less likely compared with the "foliar fertilizer
damage" hypothesis. In the discussion section, authors mentioned "*results showed a*
*remarkable negative effect of N fertilization on small-sized plants including trees,*
*understory saplings, shrubs/seedlings, and ferns. During our field investigation, we also*
*found that the average proportion of dead trees (Fig. 2d) tended to increase in N-fertilized*
*plots although the result was not statistically significant (p =0.50). Additionally, the*
*ground-cover ferns in N100 plots almost disappeared after 3.4-year N fertilization*
*(personal observation). Given the high stand density in this mature subtropical forest, we*
*suggest that N fertilization might potentially lead to increased self- and alien thinning of*
*individuals through decreasing understory light availability* (discussion 4.2 in their



paper)." However, the data provided by the Tian et al. (2017)'s experiment did not support
this idea. The canopy cover did not increase in their experiment (Table 2 in their paper),
indicating that the reduced light availability is not likely to explain the reduced understory.

Compared with the suggested mechanism above, another explanation by authors

are more plausible. By referring to the stage 3 of Aber *et al.* (1989)'s concept, authors
suggested that the decline in understory was due to the intensified N saturation: "*In our*
*experiment, the soil acidification and increased soil N concentration in high-N-fertilized*
*plots combined with the negative responses of understory plants suggest that the 3.4-year*
*N fertilization in this mature subtropical forest site has potentially caused N saturation*
(discussion 4.3 in their paper)." However, soil total N content and understory biomass
were not correlated (Fig. S1, drawn using data in Tian et al. (2017)'s Supplement),
indicating that the elevated N content in their experiment does not necessarily explain the
decrease in understory. The direct negative impact of high concentration of $NH_4NO_3$
solution seems to explain the understory decline more successfully.

In this note, I suggested a possibility of the direct negative impact of $NH_4NO_3$

application on understory vegetation. This suggestion is important because if this is the
case, the negative impact of experimental N application on understory may have been
over-estimated in several case studies using liquid $NH_4NO_3$ application (for example





Rainey *et al.*, 1999; Lu *et al.*, 2010). The prediction of the impact of elevated N deposition
on understory may be required to be re-considered.

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
