# Peer review of "Comment on "Growth responses of trees and understory plants to"

_Biogeosciences, 2017_

## Referee Comment (RC1) · Anonymous Referee #1 · 6 Sep 2017

In this commentary, Taiki Mori questioned the paper of "growth responses of trees and understory plants to nitrogen fertilization in a subtropical forest in China" by Tian et al. 2017. Generally speaking, I don't agree with the author on the questions. First, the author suggests that the reduced understory plant growth is caused by "fertilizer burn". If this is the case, there must be lots of "burned leaves" for the understory plants. However, Tian et al. didn't report any "burned leaves" in the paper. Second, the author argues "The canopy cover did not increase in their experiment, indicating that the reduced light availability is not likely to explain the reduced understory." I don't agree on this. "The canopy cover did not increase" does not necessarily lead to the unchanged "light availability". Understory vegetation itself can cause light competition.

[Figure]

Third, the author argued that "soil total N content and understory biomass were not corrected", so "the elevated N content in their experiment does not necessarily explain the decrease in understory". In Aber et al. (1989), the N saturation concept is not defined using soil total N content, but N input rate. Based on the above reasons, I don't think Mori's questions make sense.
* * *

---

## Author Comment (AC1) · 6 Sep 2017

Thank you very much for your comment. Here are my answers.

First, the author suggests that the reduced understory plant growth is caused by "fertilizer burn". If this is the case, there must be lots of "burned leaves" for the understory plants. However, Tian et al. didn't report any "burned leaves" in the paper.

Answer Thank you for your comment. But I just could not understand why you are sure there are no fertilization burn just because is was not written in the article.

Second, the author argues "The canopy cover did not increase in their experiment,

indicating that the reduced light availability is not likely to explain the reduced understory." I don't agree on this. "The canopy cover did not increase" does not necessarily lead to the unchanged "light availability". Understory vegetation itself can cause light competition.

Answer Thank you very much. I also could not understand. Do you mean understory vegetation was stimulated by N addition and created shadow? If so, why understory vegetation decreased after N addition?

Third, the author argued that "soil total N content and understory biomass were not corrected", so "the elevated N content in their experiment does not necessarily explain the decrease in understory". In Aber et al. (1989), the N saturation concept is not defined using soil total N content, but N input rate. Based on the above reasons, I don't think Mori's questions make sense

Answer Thank you very much for your comment. Yes, this hypothesis may be possible as I wrote in the main text. I just discussed that fertilization burn could explain the situation better. Since data are very limited in Tian et al. (2017)'s manuscript, I guess soil total N can be one indicator to check the author's hypothesis. If possible, please could you explain your evidence to support the author's idea? I also want to know it. Thank you.

---

## Short Comment (SC1) · 10 Sep 2017

**Nitrogen saturation and light availability influence the growth of understory plants in a nitrogen-fertilized subtropical forest: Reply to Taiki Mori, Biogeosciences Discuss., https://doi.org/10.5194/bg-2017- 358.**

Di Tian, Jingyun Fang*

Department of Ecology, College of Urban and Environmental Sciences, and Key Laboratory for Earth Surface Processes of the Ministry of Education, Peking University, Beijing, 100871, China

*Correspondence author:
Dr. Jingyun Fang
Department of Ecology, Peking University
Beijing 100871, China
E-mail: jyfang@ urban.pku.edu.cn

As atmospheric nitrogen (N) deposition has been a noteworthy aspect of global change, N fertilization experiments using ammonium nitrate ($NH_4NO_3$) are commonly applied to simulate the potential effects of N enrichment on growth of plants (e.g. H ögberg et al., 2006; Lu et al., 2010; Alvarez-Clare et al., 2013; Minocha et al., 2015) and functioning of forest ecosystems (e.g. Magill et al., 2004; Cleveland and Townsend, 2006; Mo et al., 2008; Gurmesa et al., 2016). Taking the increasing N deposition in eastern China into consideration (Liu et al., 2013), we set the dosages of N fertilization as 50 kg N ha$^{-1}$ yr$^{-1}$ (abbreviated as N50) and 100 kg N ha$^{-1}$ yr$^{-1}$ (N100) to examine potential effects of high N deposition on growth of trees and understory plants in subtropical forest ecosystems (Tian et al., 2017a). The dosages of N50 and N100 were also used in many of previous studies conducted in boreal and temperate forests in Europe and America, and tropical and subtropical forests (e.g. Rainey et al., 1999; H ögberg et al., 2006; Lu et al., 2010; Alvarez-Clare et al., 2013).

After 3.4 years' N fertilization, we found significant decreases in growth rates of small trees, understory saplings, shrubs, seedlings and ferns, and insignificant changes in those of large trees with DBH>10 cm. We proposed several possible mechanisms that regulate these different responses for trees and understory plants, such as nutrient limitation, low understory

light availability, and potential N saturation (Tian et al., 2017a). Recently, Mori (2017) commented our paper and questioned some of our explanations. Mori's comments can be summarized as two points: (1) we didn't consider the impact of 'fertilizer burn'; and (2) he cast doubt on our viewpoint about the negative effects of understory light limitation and potential N saturation.

We appreciate his comments, but wonder if he over-stated the direct impact of $NH_4NO_3$ solution on plant leaves in our experiment and ignored differences of the $NH_4NO_3$ application between forest and agricultural ecosystems. We agree that the "fertilizer burn" is common to crops with extremely high concentration of N on leaves in agricultural ecosystems (Neumann et al., 1981; Fageria et al., 2009). However, the possibility of direct exposure of leaves to $NH_4NO_3$ solution and the risk of direct foliar fertilizer damage on the understory in our experiment are very low, because our experiment was designed to explore the effects of enhanced N entering soil on plant growth and ecological functions in the subtropical forest ecosystem, in which direct leaf fertilization must be avoided to eliminate the risk of potential foliar damage.

In our experiment, the $NH_4NO_3$ solution was applied to the ground carefully by a back-hatch sprayer, but not sprayed directly onto leaves of plants (Photo 1), although spraying may exert some effects on leaves of some ground-cover plants (such as *Woodwardia japonica*). Moreover, annual precipitation and throughfall in this humid subtropical forest are quiet high (1700 and 1500 mm, respectively), which scour the plant leaves frequently and thus wipe out the risk of leaf $NH_4NO_3$ solution residual even when the ground-cover ferns are accidentally exposed to the $NH_4NO_3$ solution. As a direct evidence of negligible effects of spraying $NH_4NO_3$ solution on the leaves, Photo 2 shows clearly no significant difference in plant leaf growth performances before and after $NH_4NO_3$ solution application. Hence, the direct 'foliar fertilizer damage' raised by Mori (2007) could be hard to explain the negative effects of N fertilization on understory plants in our experiment.

Why N fertilization decreased the growth of understory plants? Light availability and soil nutrient condition were regarded as two main factors shaping growth of understory plants (e.g. Rainey et al., 1999; Dirnböck et al., 2014; Gurmesa et al., 2016; Walter et al., 2016). Accordingly, we highlighted the effects of these two factors on the growth of understory plants in our paper (Tian et al., 2017a). Here, again we would like to state some more about these two factors.

[Figure]

**Photo 1.** The application of $NH_4NO_3$ solution on the ground using a back-hatch sprayer during our experimental process.

[Figure]

**Photo 2.** Understory plants before and after $NH_4NO_3$ solution application, which were taken in July 2015 to illustrate if there are differences in leaf growth performance before and after the N application. (a) - (b) represent *Sarcandra glabra*; (c) - (d) represent *Cleyera japonica*; and (e) - (f) represent *Camellia cuspidate*.

As light availability plays a critical role in nutrient utilization and photosynthesis of understory plants in the closed forests (Strengbom and Nordin, 2008; Alvarez-Clare et al., 2013; Record et al., 2016), we took a series of canopy photos in different N treatments to illustrate if there are differences in leaf morphology and tree canopy cover among N treatments. Although the data of canopy cover estimated by weighted ellipsoidal method showed no remarkable differences, which might be resulted from the fluctuation of understory light irradiance during a day, we found tighter and denser crown of trees in the N fertilized plots during our field observation. Given the denser tree canopy in the N fertilized plots, we inferred the possibility of self- and alien thinning of individuals, especially of the small-sized plants in those plots, because of their light competition. This inference resonates with the views of previous studies that trees tended to allocate more carbon and nutrient to aboveground organs to stimulate the expansion of tree crown to gain more light resources with nutrient fertilization, which may result in reduction of light availability and difficulty of nutrient interception for understory plants (Alvarez-Clare et al., 2013; Schroth et al., 2015; Ibáñez et al., 2016).

The N saturation is suggested as another major factor influencing growth of understory plants in our paper (Tian et al., 2017a). Many previous studies have revealed that N saturation after long-term N fertilization influenced significantly plant growth and forest ecosystem functioning (e.g. Aber et al., 1998; Wallace et al., 2007; De Schrijver et al., 2007), because excess N accumulation beyond the demand of ecosystems may induce changes in soil environment (e.g. bulk density, pH, nutrient content, microbial community) and plant nutrient stoichiometric characteristics (DeHayes et al., 1999; Högberg et al., 2006; Mayor et al., 2014). For example, N application at Harvard Forest showed that N addition increased the partitioning of excess N into foliar stress-related metabolites, resulting in the higher mortality of pines (Minocha et al., 2015). With excessive N accumulated in soil, aggregated soil acidification altered the balance of base cation and micronutrient (Wang et al., 2017) and microbial community (Geisseler and Scow, 2014). In particular, the mobilization and accumulation of phytotoxic metal ions, such as aluminum ($Al^{3+}$) and manganese ($Mn^{2+}$), could suppress photosynthetic capability and survival of plants (Lu et al., 2014). Hence, plant growth in the subtropical forest in our study could be influenced by the potential N saturation that was inferred from the decreasing soil pH, increasing soil N and Mn concentrations, disturbed microbial community, and reduction of understory plant biomass following the N fertilizations (Tian et al., 2017a and 2017b). That there were no significant correlations between soil total N content and understory biomass, argued by Mori (2017), may be partly

because of the confounding effects of soil nutrient heterogeneities and low plot replications. Similar viewpoint was also claimed in a nutrient-fertilized tropical forest (Alvarez-Clare et al., 2013).

In conclusion, Mori's criticisms on our explanations for the lower understory growth under N addition are incorrect. Considering that some uncertainties remain in our study, we take this opportunity to suggest that further observations are required to explore the mechanisms underlying the changes of different growth forms with the effects of N addition in the subtropical forests. We also suggest that more attention should be paid to the plant-soil interactions to untangle the confounding effects of N fertilization and other factors on plant growth in this subtropical forest as shown in our previous studies (Tian et al., 2017a and 2017b).

**References**

Aber, J., McDowell, W., Nadelhoffer, K., Magill, A., Berntson, G., Kamakea, M., McNulty, S., Currie, W., Rustad, L., and Fernandez, I.: Nitrogen saturation in temperate forest ecosystems: hypotheses revisited, BioScience, 48, 921-934, 1998.

Alvarez-Clare, S., Mack, M.C., and Brooks, M.: A direct test of nitrogen and phosphorus limitation to net primary productivity in a lowland tropical wet forest, Ecology, 94, 1540-1551, 2013.

Cleveland, C. C., and Townsend, A. R.: Nutrient additions to a tropical rain forest drive substantial soil carbon dioxide losses to the atmosphere. P. Natl. Acad. Sci. USA, 103(27), 10316-10321, 2006.

DeHayes, D. H., Schaberg, P. G., Hawley, G. J., and Strimbeck, R. G.: Acid rain impacts on calcium nutrition and forest health: alteration of membrane-associated calcium leads to membrane destabilization and foliar injury in red spruce. BioScience, 49(10), 789-800, 1999.

De Schrijver, A., Verheyen, K., Mertens, J., Staelens, J., Wuyts, K., and Muys, B.: Nitrogen saturation and net ecosystem production. Nature, 447, 848-850, 2007.

Dirnböck, T., Grandin, U., Bernhardt-Römermann, M., Beudert, B., Canullo, R., Forsius, M., Grabner, M. T., Holmberg, M., Kleemola, S., Lundin, L., Mirtl, M., Neumann, M., Pompei, E., Salemaa, M., Starlinger, F., Staszewski, T., and Uzieblo, A. K.: Forest floor

vegetation response to nitrogen deposition in Europe. Global Change Biol., 20, 429-440, 2014.

Fageria, N.K., Barbosa Filho, M. P., Moreira, A., and Guimarães, C. M.: Foliar fertilization of crop plants. J Plant Nutrition., 32, 1044-1064, 2009.

Geisseler, D., and Scow, K. M.: Long-term effects of mineral fertilizers on soil microorganisms – A review. Soil Biol. Biochem., 75, 54-63, 2014.

Gurmesa, G. A., Lu, X. K., Gundersen, P., Mao, Q. G., Zhou, K. J., Fang, Y. T., and Mo, J. M.: High retention of $^{15}$N-labeled nitrogen deposition in a nitrogen saturated old-growth tropical forest. Global Change Biol., 22(11), 3608-3620, 2016.

Högberg, P., Fan, H. B., Quist, M., Binkley, D., and Tamm, C. O.: Tree growth and soil acidification in response to 30 years of experimental nitrogen loading on boreal forest, Global Change Biol., 12, 489-499, 2006.

Ibáñez, I., Zak, D. R., Burton, A. J., and Pregitzer, K. S. Chronic nitrogen deposition alters tree allometric relationships: implications for biomass production and carbon storage. Ecol. Appl., 26(3), 913-925, 2016.

Liu, X. J., Zhang, Y., Han, W. X., Tang, A. H., Shen, J. L., Cui, Z. L., Vitousek, P., Erisman, J. W., Goulding, K., and Christie, P.: Enhanced nitrogen deposition over China, Nature, 494, 459-462, 2013.

Lu, X. K., Mo, J. M., Gilliam, F. S., Zhou, G. Y., and Fang, Y. T.: Effects of experimental nitrogen additions on plant diversity in an old‐growth tropical forest, Global Change Biol., 16, 2688-2700, 2010.

Lu, X. K., Mao, Q. G., Gilliam, F. F., Luo, Y. Q., and Mo, J. M.: Nitrogen deposition contributes to soil acidification in tropical ecosystems. Global Change Biol., 20, 3790-3801, 2014.

Magill, A. H., Aber, J. D., Currie, W. S., Nadelhoffer, K. J., Martin, M. E., McDowell, W. H., Melillo, J. M., and Steudler, P.: Ecosystem response to 15 years of chronic nitrogen additions at the Harvard Forest LTER, Massachusetts, USA. Forest Ecol. Manag., 196, 7-28, 2004.

Mayor, J. R., Wright, S. J., and Turner, B. L.: Species-specific responses of foliar nutrients to long-term nitrogen and phosphorus additions in a lowland tropical forest. J. Ecol., 102(1), 36-44, 2014.

Minocha, R., Turlapati, S. A., Long, S., McDowell, W. H., and Minocha, S.C.: Long-term

trends of changes in pine and oak foliar nitrogen metabolism in response to chronic
nitrogen amendments at Harvard Forest, MA. Tree Physiol., 35(8), 894-909, 2015.

Mo, J. M., Zhang, W., Zhu, W. X., Gundersen, P., Fang, Y. T., Li, D. J., and Wang, H.:
Nitrogen addition reduces soil respiration in a mature tropical forest in southern China.
Global Change Biol., 14(2), 403-412, 2008.

Mori T.: Comment on "Growth responses of trees and understory plants to nitrogen
fertilization in a subtropical forest in China". Biogeosci. Discuss.,
https://doi.org/10.5194/bg-2017-358.

Neumann, P. M., Ehrenreich, Y., and Golab, Z.: Foliar fertilizer damage to corn leaves:
relation to cuticular penetration. Agron. J., 73, 979-982, 1981.

Rainey, S. M., Nadelhoffer, K. J., Silver, W. L., and Downs, M. R.: Effects of chronic nitrogen
additions on understory species in a red pine plantation, Ecol. Appl., 9, 949-957, 1999.

Record, S., Kobe, R. K., Vriesendorp, C. F., and Finley, A. O.: Seedling survival responses to
conspecific density, soil nutrients, and irradiance vary with age in a tropical forest.
Ecology, 97(9), 2406-2415, 2016.

Strengbom, J., and Nordin, A.: Commercial forest fertilization causes long-term residual
effects in ground vegetation of boreal forests. Forest Ecol. Manag., 256, 2175-2181,
2008.

Schroth, G., da Mota, M. S. S., and de Assis Elias, M. E.: Growth and nutrient accumulation
of Brazil nut trees (Bertholletia excelsa) in agroforestry at different fertilizer levels. J.
For. Res., 26(2), 347-353, 2015.

Tian, D., Li, P., Fang, W. J., Xu, J., Luo, Y. K., Yan, Z. B., Zhu, B., Wang, J. J., Xu, X. N., and
Fang, J. Y.: Growth responses of trees and understory plants to nitrogen fertilization in a
subtropical forest in China. Biogeosciences, 14, 3461-3469, 2017a.

Tian, D., Jiang, L., Ma, S. H., Schmid, B., Xu, L. C., Zhu. J. X., Li, P., Lasapio, G., Jing, X.,
Zheng, C. Y., Shen, H. H., Zhu, B., and Fang, J. Y.: Effects of nitrogen deposition on soil
microbial communities in temperate and subtropical forests in China. Sci. Total Environ.,
doi: 10.1016/j.scitotenv.2017.06.057, 2017b.

Tian, D.: Effects of nutrient fertilization on the main processes of carbon cycling in
subtropical forests. Peking University (*PhD thesis in Chinese*), 2017.

Wallace, Z. P., Lovett, G. M., Hart, J. E., and Machona, B.: Effects of nitrogen saturation on
tree growth and death in a mixed-oak forest. Forest Ecol. Manag., 243, 210-218, 2007.

Walter, C. A., Raiff, D. T., Burnham, M. B., Gilliam, F. S., Adams, M. B., and Peterjohn, W. T.:

Nitrogen fertilization interacts with light to increase Rubus spp. cover in a temperate forest. Plant Ecol., 217, 421-430, 2016.

Wang, R. Z., Dungait, A. J., Buss, H. L., Yang, S., Zhang, Y.G., Xu, Z.W., and Jiang, Y.: Base cations and micronutrients in soil aggregates as affected by enhanced nitrogen and water inputs in a semi-arid steppe grassland. Sci. Total Environ., 575, 564-572, 2017.